# Study on Preparation of Chitosan/Polyvinyl Alcohol Aerogel with Graphene−Intercalated Attapulgite(GO−ATP@CS−PVA) and Adsorption Properties of Crystal Violet Dye

**DOI:** 10.3390/nano12223931

**Published:** 2022-11-08

**Authors:** Weikai Sun, Hongxiang Ou, Ziwei Chen

**Affiliations:** School of Environmental Science and Engineering, Changzhou University, Changzhou 213164, China

**Keywords:** attapulgite, graphene oxide, chitosan, polyvinyl alcohol, crystal violet, adsorption

## Abstract

Adsorption is one of the effective methods of treating dye wastewater. However, the selection of suitable adsorbent materials is the key to treating dye wastewater. In this paper, GO−ATP was prepared by an intercalation method by inserting graphene oxide (GO) into the interlayer of alabaster attapulgite (ATP), and GO−ATP@CS−PVA aerogel was prepared by co−blending−crosslinking with chitosan (CS) and polyvinyl alcohol (PVA) for the adsorption and removal of crystalline violet dye from the solution. The physicochemical properties of the materials are characterized by various methods. The results showed that the layer spacing of the GO−ATP increased from 1.063 nm to 1.185 nm for the ATP, and the specific surface area was 187.65 m^2^·g^−1^, which was 45.7% greater than that of the ATP. The FTIR results further confirmed the success of the GO−ATP intercalation modification. The thermogravimetric analysis (TGA) results show that the aerogel has good thermal stability properties. The results of static adsorption experiments show that at 302 K and pH 9.0, the adsorption capacity of the GO−ATP@CS−PVA aerogel is 136.06 mg·g^−1^. The mass of the aerogel after adsorption−solution equilibrium is 11.4 times that of the initial mass, with excellent adsorption capacity. The quasi−secondary kinetic, Freundlich, and Temkin isotherm models can better describe the adsorption process of the aerogel. The biobased composite aerogel GO−ATP@CS−PVA has good swelling properties, a large specific surface area, easy collection and a low preparation cost. The good network structure gives it unique resilience. The incorporation of clay as a nano−filler can also improve the mechanical properties of the composite aerogel.

## 1. Introduction

Although organic dyes have a wide range of applications in different industrial fields such as textiles, leather, plastics, and printing [1], their discharge of large amounts of dye wastewater poses a serious threat to the global environment, causing pollution and endangering human health. For example, crystalline violet (CV) is a triphenylmethane organic cationic dye, which is mainly used in commercial fabric dyeing, leather processing and the food industry, but CV has been found to have harmful effects on humans and is suspected of being a potential carcinogen causing tetraplegia, kidney failure and permanent blindness; it can additionally exist in the environment for a long time [2]. Therefore, it is necessary to choose a suitable treatment method before discharging organic dye wastewater [3]. Various methods have been used to treat dye wastewater, such as photocatalytic degradation [4], membrane separation [5], advanced oxidation [6] and adsorption [7], among which adsorption is widely used for its ease of operation, simplicity and efficiency, environmental friendliness, and economic feasibility.

Great progress has been made in the development of adsorbents in recent years, but further development of adsorbents with high adsorption capacity, structural stability and easy recovery is needed to treat dye wastewater. Natural nano−clay minerals have attracted much attention because of their advantages such as a large specific surface area, non−toxicity and low cost. Attapulgite (ATP) is a chain−layered structure of magnesium−aluminosilicate clay mineral nanomaterials [8], consisting of two continuous tetrahedral layers and discontinuous octahedral layers with a unique nanorod morphology, a large surface area due to the abundant pores in its structure, and many active dots; it has been applied in the removal of organic dyes and heavy metal ions [9]. However, there are certain limitations in the practical application of ATP. On the one hand, ATP is difficult to separate from an aqueous solution in a powder state, and on the other hand, it has low selectivity due to the small ATP binding constant. Therefore, ATP needs to be modified to improve its adsorption capacity to the target.

Commonly used modification methods include heat treatment, acid modification, intercalation, and organic modification [10,11,12] to break the adsorption limitation of ATP. Among them, the intercalation method has the advantage of the ion−exchange properties of nano−clay minerals and the scalability of interlayer distances by inserting specific intercalators into nano−mineral layers to increase the layer spacing and specific surface area of the material [13]. Chen et al. prepared a new affinity membrane with good adsorption capacity and high throughput for Bisphenol A removal by β−cyclodextrin modified GO [14]. Lian et al. prepared phosphate−functionalized graphene oxide using the Arbuzov reaction, where phosphate functional groups were doped into GO by forming P−C bonds, thus improving the adsorption capacity of resorcinol [15]. The adsorption of CV by the intercalation modification of GO into ATP nanosheet layers to form a nested structure has been seldom reported. GO with good dispersion can be intercalated into the ATP, which not only prevents GO agglomeration, but also adds more reactive dots to the ATP. Layered materials with a large interlayer space and excellent properties are more favorable for the formation of intercalated composites, and Zhu et al. successfully prepared the mesoporous intercalated structures g−C_3_N_4_@ATP by eutectic technique and calcination, which provided nanoscale space for photocatalytic reactions, thus improving the thermal and molecular collision efficiency [16]. Li et al. used the quaternary ammonium surfactant dodecyl benzyl ammonium chloride to modify montmorillonite by intercalation, and the layer spacing was increased from 1.53 nm to 2.97 nm, which improved the removal of antibiotics [17]. By inserting layers, the space inside the material is improved, which is more conducive to the chemical reactions. Therefore, the construction of the GO intercalation layer of the ATP is an effective way to improve the adsorption performance of the material.

To facilitate material recycling, the natural polysaccharide biomaterial chitosan (CS) was added, which has the advantages of non−toxicity, biocompatibility, and low cost [18,19] and has a wide range of applications in drug delivery systems, food packaging, and contaminant adsorption. The molecular chains of CS contain active functional groups such as amino and hydroxyl groups, which not only improve the mechanical properties of the material, but also have excellent ion adsorption properties [20]. At the same time, CS can be used as a carrier for ATP immobilization to prepare aerogels of different shapes so that the material has a certain hardness and shape, solving the problem that nano−powder materials are hard to recover and increasing the adsorption capacity by further modifying the synergistic effect of the ATP. To improve the mechanical properties of aerogels polyvinyl alcohol (PVA), a highly hydrophilic, biocompatible polymer with low toxicity, good mechanical properties, and a high molecular structure [21], was also incorporated to give aerogel materials good flexibility for better removal of heavy metals and dyes from wastewater.

In this study, the GO−ATP was prepared by the intercalation method, and the GO−ATP@CS−PVA composite aerogel was subsequently prepared by co−blending−crosslinking for the removal of crystalline violet dye from the solution. Different characterization methods were used to investigate the physicochemical properties of the aerogel adsorbent materials. The effects of the GO−ATP@CS−PVA on the removal of crystalline violet dye from solutions were investigated by various adsorption experiments.

## 2. Materials and Methods

### 2.1. Materials

Attapulgite (ATP) from Xuyi, Jiangsu, China, 200 was mesh screened. Sodium hexametaphosphate ((NaPO_3_)_6_), graphene oxide (GO), polyvinyl alcohol (PVA) and glutaraldehyde were all analytically pure when purchased from Sinopharm Chemical Products, Co., LTD, Shanghai, China. Crystalline violet (CV) and chitosan (CS) were purchased from Aladdin as analytical purity, and deionized water was used throughout the experiments.

### 2.2. Experimental Instruments

Scanning electron microscope (SEM, JSM−7001F); Thermogravimetric analyzer (TGA, NETZSCH TG−209−F3); Specific surface and pore size analyzer (ASAP2010C); Fourier transform infrared spectrometer (FT−IR, Nicolet NEXUS−470); Ultraviolet spectrophotometer (UV−2450); Microcomputer−controlled electronic universal testing machine (100 KN, UTM5105).

### 2.3. GO Intercalation Modified ATP

First, 5 g of the ATP was weighed into a 5% hydrochloric acid solution with a solid–liquid mass ratio of 1:5, stirred, sonicated for 2 h, centrifuged and washed with deionized water to neutral, dried in an oven at 60 °C for 24 h, removed and ground and sieved. Then, 3 g of pretreated ATP and 300 mg of GO were ultrasonically dispersed in deionized water, and the brown GO suspension and the aqueous solution of the ATP dispersion were mixed well; 200 mg of (NaPO_3_)_6_ was then added at 70 °C with continuous stirring and ultrasonication for 3 h to ensure that the GO and ATP were intermixed and intercalated, centrifuged at 12,000 rpm and washed several times with deionized water, before being dried in an oven at 60 °C for 24 h. The final product graphene oxide intercalated gravimetric composite (GO−ATP) is shown in Figure 1.

### 2.4. Preparation of GO−ATP@CS−PVA Composite Aerogel

First, 2 g of CS in 50 mL of aqueous 2% acetic acid solution were dissolved and stirred well in a water bath at 60 °C. Then, 3 g of GO−ATP was placed in a beaker, deionized water was added to the sonicate to disperse it well, and was slowly added to the CS solution and mixed well. Forty mL of 5% PVA aqueous solution was obtained in a water bath at 95 °C. After cooling, it was added to the homogeneous mixed solution and mixed and stirred well and sonicated to defoam; then, 5% glutaraldehyde aqueous solution was added slowly dropwise for chemical cross−linking, continuously stirred well and then poured into the mold and left until its complete gelation. Finally, it was freeze−dried in a vacuum at −50 °C for 48 h to obtain GO−ATP@CS−PVA composite aerogel.

### 2.5. Swelling Experiments

The swelling performance of the GO−ATP@CS−PVA in pH 6.5 deionized water at 25 °C was studied using the weight method. A hundred mg dry GO−ATP@CS−PVA aerogel was placed into the deionized solution, and taken out after 5, 10, 20, 30, 60, 120, 180, 360 min, respectively. The water was then wiped off the surface of the samples with filter paper and the wet weight of the samples was weighed. The swelling rate (SR) [22] was calculated using Equation (1) until the adsorbent no longer swelled, and the swelling equilibration time was recorded.
(1)$$SR\%=\frac{W_{s}-W_{d}}{W_{d}}\times 100\%\;$$
where *W_s_* and *W_d_* are the weights of the swollen and dried samples, respectively.

### 2.6. Adsorption Experiments

Adsorption experiments were carried out to investigate the effects of the initial pH of the solution, adsorbent dosage, adsorption time, initial CV concentration and temperature at the time of the CV removal. To study the solution pH on the adsorption effect, 50 mg of GO−ATP@CS−PVA was injected into 50 mL of CV dye solution with a concentration of 200 mg·L^−1^ at pH 3.0, 5.0, 7.0, 9.0 and 11.0, respectively, and the adsorption time was 360 min at room temperature. In the material dosing experiments, 10, 20, 30, 40 and 50 mg of GO−ATP@CS−PVA were added to 50 mL of the CV dye solution with a concentration of 200 mg·L^−1^ at pH 9.0, and the adsorption time was 360 min at room temperature to select the optimal dosing amount. The adsorption kinetics experiments were carried out by adding 20 mg of adsorbent material to 20 mL of the CV dye solution at pH 9.0 with a concentration of 200 mg·L^−1^ and studying the adsorption equilibrium time after adsorption at 303 K for 5, 10, 20, 30, 60, 120, 180, 360, 720 and 1440 min at room temperature, respectively. To observe the effects of temperature on the experiment, adsorption experiments were carried out at three different temperatures of 292 K, 297 K and 302 K. Twenty mg of the material were added to 20 mL of solutions with different CV concentrations (50 mg·L^−1^, 100 mg·L^−1^, 150 mg·L^−1^, 200 mg·L^−1^, 250 mg·L^−1^) at pH 9.0 for 360 min. After adsorption, the adsorbent was separated from the solution, and the adsorbed solution was aspirated and the concentration of the CV was measured at 590 nm by a UV spectrophotometer (UV−2450, Shimadzu, Japan). Equation (1) for the adsorption amount (*q_e_*, mg·g^−1^) is as follows.
(2)$$q_{e}=\frac{\left(C_{0}-C_{e}\right)V}{m}$$
where *C_o_* (mg·L^−1^) and *C_e_* (mg·L^−1^) are the initial and equilibrium concentrations of the CV in the solution, respectively; V (mL) is the volume of the solution; and m (g) is the mass of the adsorbent.

## 3. Results

### 3.1. Characterization Results and Analysis

By observing the scanning electron microscope (SEM) in Figure 2a, it can be seen that the microscopic morphology of the ATP is strip−shaped, rod−shaped or fiber−lamellar, and the rod−shaped single crystals are closely arranged, and most of them are aggregated rod−shaped crystal layers, which provides a good environment for intercalation modification. In Figure 2b, the GO sheets are stacked and folded, which provides more contact area for adsorption, thus helping to achieve a better adsorption effect. From Figure 2c, it can be seen that the GO was successfully loaded onto the ATP nano−sheet, and there were even and good holes in the material, showing a honeycomb structure. The GO was well dispersed in the ATP matrix, which changed the interlayer spacing of the material and increased the specific surface area of the material. It can be seen from Figure 2d that the GO−ATP@CS−PVA shows the typical porous structure of outgassing gel, revealing a three−dimensional network structure, and the rough wrinkled surface will be favorable for loading more dye molecules.

XRD and FT−IR analysis results are shown in Figure 3a,b. According to Figure 3a, the GO diffraction peak is at 2θ = 12.02° and ATP exhibits a major peak at 8.32° corresponding to its crystal plane. For GO−ATP composites, the diffraction peaks are significantly shifted toward the lower 2θ position, exhibiting relatively strong diffraction at 7.46°, and the interlayer spacing d is calculated using the Bragg equation (2 dsinθ = nλ) [23] to gradually increase from approximately 1.063 nm for the pristine ATP to 1.185 nm for the GO−ATP, and the change in the layer spacing confirms the successful intercalation of the GO into the ATP nanosheet layers. In Figure 3b, the GO presents contraction vibration peaks at 1076, 1223, 1650, and 1721 cm^−1^ corresponding to O−H, C−O−C, C=C, and C=O stretching vibrations, respectively [24]. The FT−IR of the ATP shows main adsorption bands of 3617, 3418 and 1657 cm^−1^ which are associated with the stretching and bending vibrations of −OH and at the peak 1031 cm^−1^ is the stretching vibration of the Si−O in the ATP framework. The vibration band at 1461 cm^−1^ belongs to the contraction vibration of the −OH group, and the peak position of the GO−ATP after intercalation modification is slightly toward 1657 cm^−1^, which is due to the condensation reaction of the GO carboxyl group in the process of intercalation and consumption of the hydroxyl group on the ATP. Compared with the peaks of the original GO and ATP, the GO−ATP composites become weaker at 1657 cm^−1^ for the oxygen−containing characteristic peak and the peak at 1031 cm^−1^ becomes denser, forming a new vibrational peak C−Si−O [25], which also implies that the GO successfully becomes intercalated into the modified ATP.

From the nitrogen adsorption desorption curves and pore size distribution curves of the materials in Figure 3c,d, it can be seen that the adsorption isotherms of both the ATP and GO−ATP are type IV adsorption isotherms [26], with a smooth and then rising trend in the first phase of the ATP, a slow rising trend in the medium pressure curve, and a rapid rise in the relative pressure in the range of 0.8–1.0, indicating the presence of microporous structures, belonging to the H3 type of hysteresis loop isotherm. The adsorption isotherms of the GO−ATP composites have an H4 type hysteresis loop, which may be due to the effective dispersion of the GO onto the ATP lamellae, forming micro−mesopores and a laminar aggregated structure with a high specific surface area [27]. The specific surface area of the ATP was 128.78 m^2^·g^−1^, while the intercalated GO−ATP increased its specific surface area to 187.65 m^2^·g^−1^. The pore size of the GO−ATP was mainly distributed at 2–12 nm, with an average pore size of 8.41 nm.

In Figure 4a, for the PVA the FT−IR profile at 3450 cm^−1^ is attributed to the absorption peaks at O−H stretching vibrations, which are caused by strong hydrogen bonding interactions between the hydroxyl groups; the absorption peaks at 2940, 1720 cm^−1^ stretch the vibrations of the C−H and C=O vibrational absorption peaks in the PVA molecular chain [28]. The presented absorption peaks of the CS IR spectra at 3342, 1523 cm^−1^ are overlapped by N−H stretching vibrations and at 1657 cm^−1^ are caused by C=O stretching vibrations, while the CS absorption peak in the GO−ATP@CS−PVA shifts from 3342 cm^−1^ to a higher wave number, 3430 cm^−1^, due to the protonation of the amino group as a result of weaker hydrogen bonding interactions during the dissolution of the CS in acetic acid [29]. The peak stretching vibrations of O−H, C−H and C=O were all shifted to higher frequencies after the addition of the PVA, which indicated strong interactions between the hydroxyl group (−OH) and the amino group (−NH_2_) of the CS after the addition of the PVA, and implied that the GO−ATP@CS−PVA aerogel had been successfully prepared.

Figure 4b shows the thermogravimetric analysis graphs of the GO−ATP@CS and GO−ATP@CS−PVA. Thermogravimetric analysis can study the thermal stability of materials, the decomposition process of substances and other chemical phenomena. The materials were subjected to a temperature rise rate of 20 °C min^−1^ and 40–820 °C. The weight loss was faster when reaching 120 °C. The initial weight loss was physical adsorption water because the abundant hydroxyl and amino structures can absorb large amounts of water; the weight loss at 150–400 °C was caused by the decomposition of the PVA chains and CS network structure [30]. Further thermal degradation at 400–800 °C can be attributed to the PVA backbone and inorganic compounds. It can be seen in the figure that the addition of the PVA improves the thermal stability of the GO−ATP@CS−PVA aerogel, and that the material has 52.6% of residual material mass left after 800 °C.

The compression and release process of the GO−ATP@CS−PVA aerogel was tested with a universal testing machine as shown in Figure 5a. The initial thickness of the aerogel was 8.25 mm and the pressure was released after compressing 5 mm (at 60.6% compression strain), while the thickness of the aerogel recovered to 7.52 mm with good mechanical properties measured after 5 min. Figure 5b shows the stress–strain curves of the GO−ATP@CS−PVA at a compression rate of 1 mm/min. No plastic deformation of the aerogel occurred at a compression strain of 60.6%, and the aerogel recovered rapidly after removal of the external pressure, attributed to its good network structure [31], which gives the aerogel a unique resilience; the aerogel additionally has excellent mechanical properties.

The swelling properties are an important factor affecting the utility of aerogels in the field of adsorption [32]. The swelling experiment results of the GO−ATP@CS−PVA are shown in Figure 5c. The GO−ATP@CS−PVA aerogel was rapidly adsorbed and swollen at 79.3% during the first 10 min. The water storage capacity of the aerogel reached saturation at 3 h. The swelling rate of the material reached 103.7% at the dissolution equilibrium, the water storage mass was 11.4 times the initial mass, and the volume became significantly larger, which was attributed to the hydrophilic group −OH of the PVA molecular chain and the −NH_2_ functional group on the CS. This resulted in the excellent water adsorption capacity of the GO−ATP@CS−PVA and increased the affinity of the GO−ATP@CS−PVA for water molecules. The cross−linked network structure provides a favorable spatial location for water molecule adsorption [33], and increases the water storage capacity. The fast water adsorption kinetics properties of GO−ATP@CS−PVA aerogels can be attributed to their excellent adsorption capacity.

The zeta potential of the aerogel at different pH values is a key parameter affecting the interaction between the aerogel and adsorbents [34]. As shown in Figure 6a, the amount of charge increases with the increase in pH. When the pH value of the solution is less than 5, the adsorption capacity of the material to the CV is relatively low. Under strong acidic conditions, a large amount of H^+^ in the system competes with the CV for adsorption sites on the adsorption materials, resulting in reduced adsorption capacity. When the pH of the solution is greater than 5, the GO−ATP@CS−PVA becomes negatively charged on the surface because of the presence of carboxyl and silicate groups in its chemical structure [35]. Therefore, the composite aerogel is more conducive to adsorption of the CV with a positive charge in the alkaline solution.

The pH value of the solution is an important factor affecting the absorption swelling capacity of materials. As shown in Figure 6b, the swelling capacity of the GO−ATP@CS−PVA aerogel was investigated at different pH values ranging from 3.0 to 11.0. The swelling rate of the adsorbent does not change much in the range of pH 5.0–11.0, and drops to 600.45% at pH 3.0. The decrease in the swelling rate is attributed to the protonation of the composite, which makes it difficult for the adsorbent to form hydrogen bonds with water molecules [36]. When the pH is 9.0, the swelling rate reaches 1239.31%. The excellent water adsorption ability of the aerogel can be attributed to the fact that the PVA contains a large amount of the hydroxyl group, which improves the affinity for water molecules. The internal cross−linking network structure of the material provides more favorable space for the complexation of water molecules, thus increasing the water storage capacity.

### 3.2. Adsorption Performance

#### 3.2.1. Effect of Initial pH on Adsorption

In the adsorption process, the pH values of solutions affect the chemical properties of the adsorbent and the functional groups on the surface charge of the adsorbent. In this experiment, the adsorption performance of the GO−ATP@CS−PVA aerogels on the CV under different pH conditions was investigated. As seen in Figure 7a, CV adsorption is low under strongly acidic conditions, due to the competition between the high concentration of H^+^ in the adsorption and the cationic dye [37]. With the increase in the pH value, the surface of the GO−ATP@CS−PVA aerogel material was deprotonated, and more active dots appeared on the adsorbent surface, which promoted the electrostatic attraction between the material and dye molecules; the adsorption quantity *q_e_* of the CV removal in the solution also increased. The maximum adsorption amount of the material was 136.58 mg·g^−1^ at pH 9.0.

#### 3.2.2. Effect of Material Dosing on Adsorption Experiments

The adsorption effect of the material usually grows with the increase in the adsorbent dosage and adsorption time, and the removal rate reaches a maximum when the adsorbent dosage and adsorption time reach a certain value, at which time a further increase in the adsorbent dosage and adsorption time will not enhance the adsorption effect significantly and the adsorption amount will also decrease [38]. The effect on CV adsorption was investigated by varying the adsorbent dosage, and Figure 7b shows that as the adsorbent dosage increased from 0.2 to 1.0 g·L^−1^, the CV adsorption rate increased from 7.8% to 71.5%. When the initial concentration is constant, increasing the amount of adsorbent increases the total surface area adsorbed as well as the number of active dots and the adsorption rate. The adsorption capacity of the material was 138.28 mg·g^−1^ at a dosage of 0.8 g·L^−1^, and the adsorption capacity did not change much when the adsorbent dosage was continued, increasing to 1.0 g·L^−1^. The adsorption capacity decreased due to a further increase in the adsorbent dosage resulting in the increase in active dots and the decrease in dye molecules.

#### 3.2.3. Adsorption Kinetics Experiments

Adsorption kinetics allow us to describe the adsorption rate of adsorbent materials and to study their adsorption mechanism [39]. This is not only beneficial to study the mechanism of dye removal but also to control the reaction time. The results of kinetic adsorption experiments were fitted using quasi−primary and quasi−secondary adsorption kinetics models [40], as shown in Equations (3) and (4), respectively, and the relevant parameters are shown in Table 1.
(3)$$\ln\left(q_{e}-q_{t}\right)=\ln q_{e}-k_{1}t$$
(4)$$\frac{1}{q_{e}-q_{t}}=1/q_{e}+k_{2}t$$
where *q_t_* (mg·g^−1^) is the adsorption at time *t* and *q_e_* (mg·g^−1^) is the equilibrium adsorption, which are the adsorption and equilibrium adsorption of the GO−ATP@CS−PVA on the CV at time *t*, respectively. *k*_1_ and *k*_2_ are the quasi−primary and quasi−secondary model rate constants, respectively.

Figure 7c shows that the adsorption process is very rapid in the first 60 min, and the *q_t_* increases rapidly. Then the adsorption rate gradually slows down and the adsorption reaches equilibrium after 360 min due to the decreasing binding dots of the adsorbent to adsorb the dye, and the *q_t_* remains almost constant, which means that the adsorption process has reached dynamic equilibrium. As shown in Table 1, the *R*^2^ = 0.970 of the quasi−secondary model can better fit the adsorption process compared with the quasi−first−order model, and the CV adsorption onto the GO−ATP@CS−PVA is a chemisorption−based adsorption process.

#### 3.2.4. Adsorption Isotherm Experiment

Isothermal adsorption studies the effect of adsorbent concentration on the adsorption performance of adsorbents under certain temperature conditions. The Langmuir equation and the Freundlich equation [41] are used to demonstrate the adsorption isotherm in this study. The Equations are given in (5) and (6).
(5)$$q_{e}=\frac{K_{L}q_{m}C_{e}}{1+K_{L}C_{e}}$$
(6)$$\;q_{e}=K_{F}C_{e}^{1/n}\;$$
where *q_e_* (mg·g^−1^) is the equilibrium adsorption capacity of the GO−ATP@CS−PVA; *K_L_* is Langmuir’s affinity constant, *K_F_* is Freundlich’s orientation constant; *q_m_* (mg·g^−1^) is the maximum adsorption capacity; and *C_e_* (mg·g^−1^) is the remaining concentration of the CV in the solution.

As can be seen from Figure 7d, the adsorption amounts increased with the initial concentration of the CV, and the rising trend of the adsorption amounts gradually leveled off after reaching a certain concentration. However, with the increase in the temperature, the CV adsorption amount onto the GO−ATP@CS−PVA was increased, and the adsorption of the crystalline violet by the GO−ATP@CS−PVA was a heat absorption reaction. This indicates that the adsorption process is favored under relatively high solution−temperature conditions.

As can be seen from Table 2, the equation of the Freundlich model has a higher *R*^2^ at different temperatures compared to the Langmuir model [42], and the adsorption process of the GO−ATP@CS−PVA for CV adsorption is more in line with the Freundlich model, which is mainly based on multilayer chemisorption. The maximum adsorption amount of the material *q_e_* at 302 K is 136.06 mg·g^−1^, and the *R_L_* values also ranged from 0 to 1, which were favorable for the adsorption of the CV dye by GO−ATP@CS−PVA.

The linear equation of the Temkin model is shown in (7). The Temkin isotherm in Figure 8a explains the interaction between the GO−ATP@CS−PVA aerogel and the dye molecules and describes the adsorption on heterogeneous surfaces of the material [43]. R is the gas constant of 8.314 J·mol^−1^·K^−1^, T (K) is the absolute temperature, and the equilibrium binding constant is related to the maximum binding energy and the absolute temperature [44]. According to the Temkin isotherm model simulation, it is found that the constant *A_T_* is 18.519 L·mg^−1^, *b_T_* is 0.023 kJ·mol^−1^, and the correlation coefficient is 0.8561 in the 302 K adsorption isotherm. The interaction between the adsorbent surface and CV dye molecules is weak.
(7)$$q_{e}=\frac{RT}{b_{T}}lnA_{T}+\frac{RT}{b_{T}}lnC_{e}$$

#### 3.2.5. Adsorption Regeneration Experiment

The regeneration experiment is an important aspect of studying the stability and recyclability of adsorbents [45]. In this experiment, 50 mg of material was added to 50 mL of the CV dye solution with pH 9 at a concentration of 200 mg·L^−1^ for five consecutive cycles of adsorption experiments to evaluate the regeneration performance of the GO−ATP@CS−PVA aerogel. The recovered composite aerogel was immersed in 0.01 M HCl and 0.01 M NaOH, washed several times with deionized water, and then freeze−dried for the next adsorption test. The morphology of the composite aerogel after adsorption was slightly expanded compared with that before adsorption, and the surface of the material was not damaged, showing excellent swelling performance. In Figure 8b, after five sorption−desorption cycles of the aerogel, the adsorption amount of the CV by the GO−ATP@CS−PVA was 78.4% of the initial one. The decrease in the amount of adsorption may be due to the loss of the active site of the adsorbent and incomplete desorption of the adsorbent during regeneration. The composite aerogel showed good recycling performance.

## 4. Discussion

ATP has abundant storage capacity and is used in many fields. The layered aluminosilicate structure is more conducive to the success of intercalation modification, which improves the adsorption performance in wastewater treatment and opens up new application prospects. In this context, the present work increases the layer spacing of the material by the GO intercalation of modified ATP and improves the adsorption performance of the material. Considering the poor recoverability of the nano−powder material, the aerogel with excellent mechanical properties was prepared by cross−linking the intercalation material with the CS and PVA, and the good adsorption and swelling properties of the material also laid a solid foundation for the successful development of the adsorption experiments. Putri et al. [46] prepared a biosorbent consisting of lemongrass leaf fibers combined with cellulose acetate, which showed a maximum adsorption capacity of 36.10 mg·g^−1^ for CV. AbdEl et al. [47] loaded silver nanoparticles onto activated carbon for CV removal from aqueous solutions with a maximum adsorption capacity of 87.20 mg·g^−1^. The maximum adsorption of CV dyes by GO−ATP@CS−PVA aerogel was compared with the data reported in a previous similar study, and the adsorption efficiency of this gel was superior to some previous studies and has potential applications in dye wastewater treatment.

## 5. Conclusions

In this study, the ATP was first modified by GO intercalation to increase the layer spacing of the material, and then crosslinked with the CS and PVA to prepare the GO−ATP@CS−PVA aerogel for CV dye removal. The characterization results show that the aerogel has good swelling properties, a large specific surface area and is easy to collect with a low preparation cost, while improving the adsorption performance of the material. At 60.6% compressive strain after the removal of external pressure, the GO−ATP@CS−PVA aerogel can recover rapidly. The good network structure gives the aerogel unique resilience and good mechanical properties. The results of the swelling experiment showed that the water storage capacity of the aerogel reached saturation after 3 h, and the swelling rate reached 103.7% at the swelling equilibrium, while the water storage mass was 11.4 times the initial mass. The results of the adsorption experiments showed that the Freundlich and Temkin models, as well as the quasi−secondary kinetic model were able to fit the adsorption process of the GO−ATP@CS−PVA aerogel on the crystalline violet of the cationic dye effectively, and the adsorption was an exothermic process, while the material adsorption was mainly a chemical multilayer adsorption process. Under the experimental conditions, the maximum adsorption capacity of the GO−ATP@CS−PVA aerogel was 138.28 mg·g^−1^. The adsorbent has a stable adsorption performance and has potential application prospects in wastewater treatment. This provides a new experimental idea for the development of environment−friendly dye wastewater adsorption materials.

## Figures and Tables

**Figure 1 nanomaterials-12-03931-f001:**
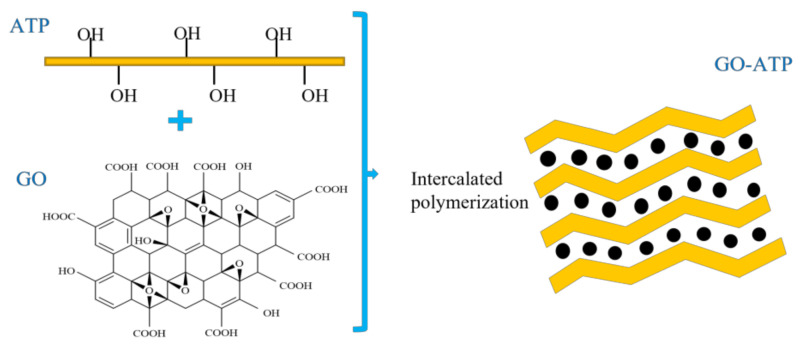
Preparation process and structure of GO−ATP.

**Figure 2 nanomaterials-12-03931-f002:**
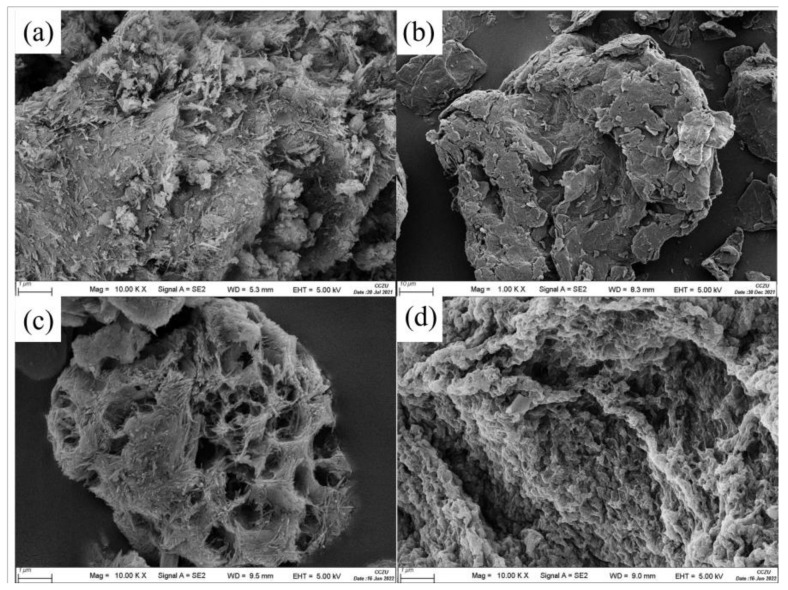
SEM images of ATP (**a**), GO (**b**), GO−ATP (**c**) and GO−ATP@CS−PVA (**d**).

**Figure 3 nanomaterials-12-03931-f003:**
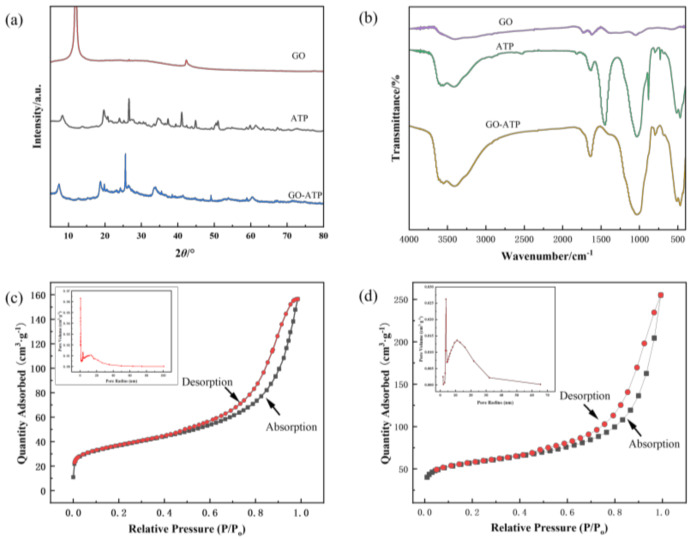
(**a**) XRD patterns of GO, ATP and GO−ATP samples; (**b**) FT−IR spectra of GO, ATP and GO−ATP composites; nitrogen adsorption and desorption isotherms and pore size distribution of ATP (**c**) and GO−ATP (**d**) specific surface area of ATP.

**Figure 4 nanomaterials-12-03931-f004:**
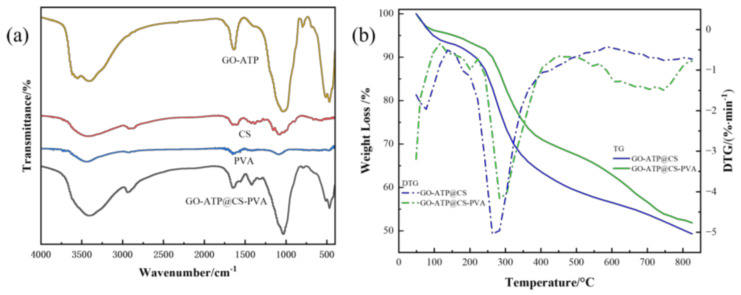
(**a**) FT−IR spectra of GO−ATP, CS, PVA and GATP@CS−PVA composites; (**b**) The TG/DTG curves of GO−ATP@CS and GO−ATP@CS−PVA.

**Figure 5 nanomaterials-12-03931-f005:**
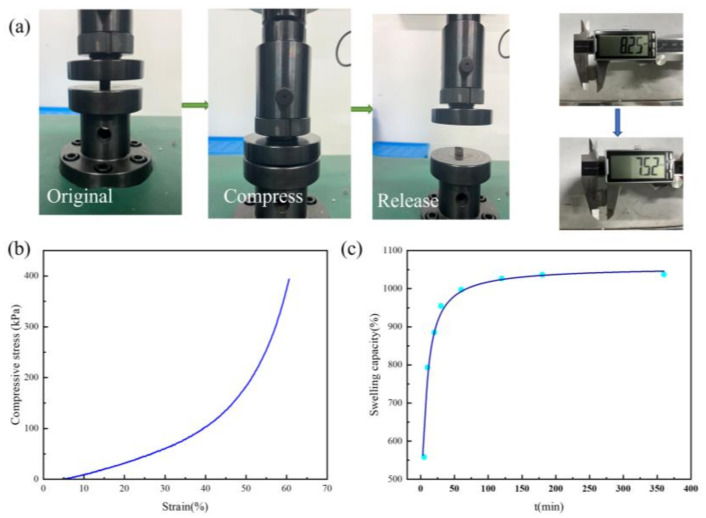
(**a**) GO−ATP@CS−PVA compression and release of aerogel; (**b**) Stress–strain curve of GO−ATP@CS−PVA; (**c**) Aerogel swelling performance test.

**Figure 6 nanomaterials-12-03931-f006:**
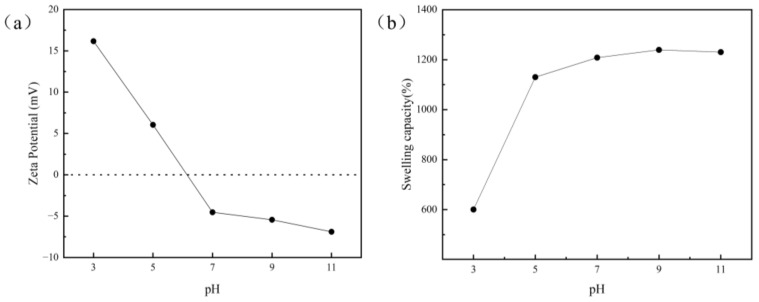
(**a**) Zeta potential of GO−ATP@CS−PVA at different pH values; (**b**) Effect of pH on GO−ATP@CS−PVA swelling capacity.

**Figure 7 nanomaterials-12-03931-f007:**
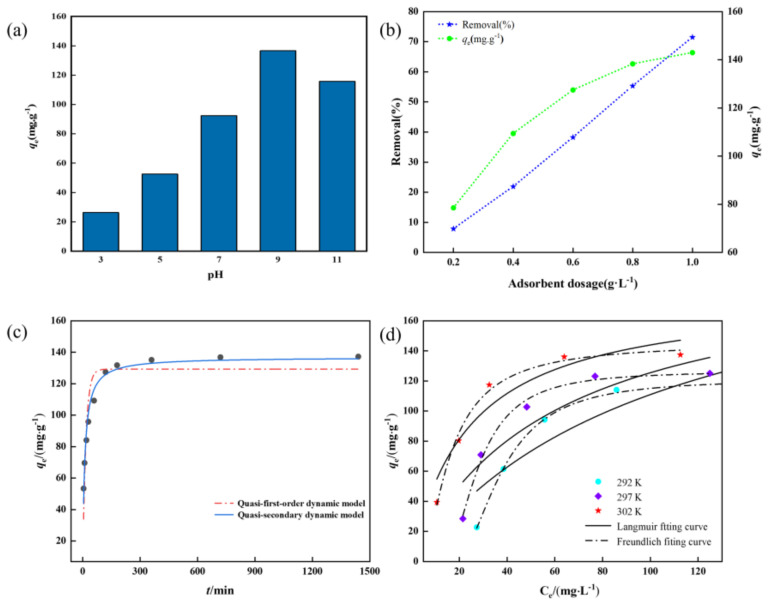
(**a**) Effect of initial pH value towards CV removal by GO−ATP@CS−PVA; (**b**) Effect of dosage on CV removal; (**c**) Kinetic model fitting curve of CV removal by GO−ATP@CS−PVA; (**d**) Isotherm model fitting of CV adsorption.

**Figure 8 nanomaterials-12-03931-f008:**
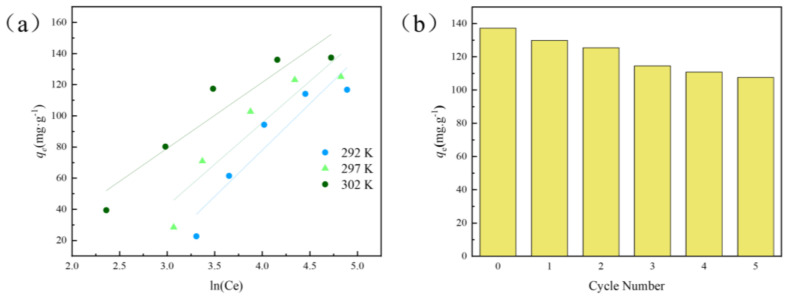
(**a**) Temkin isothermal model of GO−ATP@CS−PVA; (**b**) GO−ATP@CS−PVA adsorption regeneration experiments with CV removal.

**Table 1 nanomaterials-12-03931-t001:** Correlation coefficient of kinetic fitting.

*T*/K	*q_e_*/(mg·g^−1^) Expermental	Quasi−First Order Kinetic Equation			Quasi−Second Order Kinetic Equation		
		*q_e_*/(mg·g^−1^) Theoretical	*K*_1_/min^−1^	*R* ^2^	*q_e_*/(mg·g^−1^) Theoretical	*K*_2_/(g·mg^−1^·min^−1^)	*R* ^2^
303	137.24	129.284	0.061	0.849	136.908	0.094	0.970

**Table 2 nanomaterials-12-03931-t002:** Isotherm fitting results.

*T*/K	Langmuir				Freundlich		
	*q_m_*/(mg·g^−1^)	*K*_L_/(L·mg^−1^)	*R* _L_	*R* ^2^	*K*_F_/(L·mg^−1^)	1/*n*	*R* ^2^
292	120.834	0.158	0.076	0.774	2.987	0.158	0.998
297	127.432	0.169	0.084	0.795	4.373	0.226	0.985
302	136.057	0.184	0.095	0.890	8.259	0.435	0.984

## Data Availability

The data presented in this study are available on request from the corresponding author.

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
