# Peer review of "Study on Preparation of Chitosan/Polyvinyl Alcohol Aerogel with Graphene−Intercalated Attapulgite(GO−ATP@CS−PVA) and Adsorption Properties of Crystal Violet Dye"

_nanomaterials, 2022, doi:10.3390/nano12223931_

Round 1

Reviewer 1 Report

This manuscript relates some aspects about the removal of crystal violet from waste waters and could be accepted for publication after a minor revision.

1. The authors used polypropylene alcohol or polyvinyl alcohol?

2. English need to be improved.

3.The second phrase form section 2.3 is too long and ambiguous.

4. The notation Qe must be qe.

5. The authors have to change the word "dots" with "sites". Is more appropriate in this context.

6. I recommend investigating the reusability of the sorbent.

7. The adsorption is a spontaneous process?

8. Why this sorbent is better than others?

Author Response

Thanks for your precious suggestions. We have revised the manuscript according to your comments. The specific modifications are followed.

Reviewer 2 Report

The paper titled “Study on removal of dye crystal violet by graphene oxide intecalated attapulgite aerogel composites GO-ATP@CS-PVA” deals on the synthesis of GO-ATP composites for dye adsorption.  The manuscript is not organized and incomplete, so major mandatory revisions are requested for possible reconsideration in this journal.

1)      In the abstract authors should cite the novelty of this work

2)      The introduction is poor and more other works on modified GO for adsorption applications should be discussed.

3)      The quality of the figure 1 could be improved

4)      What is the role of ATP in the prepared composite? please explain

5)      How the CS in immobilized over the GO-ATP? Please explain the chemical phenomenon

6)      Regarding the SEM images, the discussion about this analysis is not clear and the Figure 2 should be replaced by another visible figure with the same scale.  

7)      Figure 3 is not clear, and then I cannot examine the text corresponding this figure!

8)      All the figures are in low quality and must be improved.

9)      Where is the conclusion

10)   More references should be added to confirm the results and discussion obtained for the dye adsorption

Author Response

(The authors gave the same response as above.)

Reviewer 3 Report

In the paper titled: “Study on removal of dye crystal violet by graphene oxide intercalated attapulgite aerogel composites GO-ATP@CS-PVA”, the main subject of the manuscript is the description of the adsorption performance of the GO-ATP@CS-PVA for the crystal violet dye. The synthesis of GO-ATP@CS-PVA aerogel was performed by intercalations of GO-ATP in the co-blending-crosslinking reaction of chitosan (CS) and polyvinylalcohol (PVA).

1. I suggest that the authors will take out from the title the abbreviation of the aerogel composites. They can introduce the abbreviation of the aerogel composites “GO-ATP@CS-PVA” late on in the abstract.

2. Please change in the abstract “polypropylene alcohol (PVA)” to polyvinylalcohol.

3. All Figures have a bad resolution, being very difficult to read something inside the graphic.

4. Taking into account that the authors are discussing the influence of the pH on the adsorption performance, I suggest that they measure the zeta potential for the GO-ATP@CS-PVA at different pH?

5. I am very sure that the morphology of the aerogel composites is changing together with the pH. Can the authors show the pH influence on the swelling capacity of the aerogel composites?

6. As we know, for the equilibrium adsorption studies for a pollutant, we can fit the experimental data on both the linear and non-linear forms for more isotherm models as Temkin, Dubinin-Radushkevich, Khan, etc., not only for Freundlich and Langmuir. Can the author try to fit the experimental data also for other isotherms, or did they do that?

7. Would it be nice if the manuscript also included a reusable study of the adsorption performance for the aerogel composites on the crystal violet dye? Considering the high impact factor of the journal, adding a regeneration study of the GO-ATP@CS-PVA would add value to the manuscript.

Author Response

(The authors gave the same response as above.)

Round 2

Reviewer 2 Report

In the revised manuscript, authors changed the title and added some details in different section of the manuscript.

Despite this revision I suggests major mandatory revision before publication of this work.

1)      The abstract is not organized and should be rewritten and shortened. In its actual forma, authors added some line at the end of the abstract or its not accepted to repeat some results already present! For example: In this study, the bio-based aerogel was prepared, and graphene-intercalated attapulgite was introduced into CS/PVA polymer matrix ….. While these lines contain the same sentence of the GO-ATP material was prepared by intercalation of graphene oxide (GO) into attapulgite (ATP), and GO-ATP@CS-PVA aerogel was synthesized by blending crosslinking with chitosan (CS) and polyvinyl alcohol (PVA). GO-ATP@CS-PVA aerogel was used to aerogel. So please reformulate the abstract by rewriting it according the obtained results.

2)      Polyvinyl alcohol (PVA) should be added in the keywords list

3)      In my last queries I have asked the authors to add and discuss more articles on the modification of GO to explore the subject and novelty of the present work, while the authors did not revise the introduction?

4)      A comparison on the adsorption of Crystal violet dye should be added to show the originality and the effectiveness of the prepared materials

5)      Authors said that “According to the template of the journal, we include the conclusion in the discussion section.” While when I reed the recent published article I the journal I see that all paper contain conclusion, separately. I suggest the structure (  1) Introduction, 2) Materials and Methods, 3) Results, 4) Discussion and 5) Conclusion  ).    So conclusion could be added separately to the Discussion part.

Author Response

(The authors gave the same response as above.)

Reviewer 3 Report

Now I agree with the manuscript of this paper in its present form.

Author Response

Thank you very much for your approval.

Round 3

Reviewer 2 Report

Minor revision:

1) Table 1 should be revised

2) The first line of conclusions: "The study first increased the layer spacing of the material by GO" ? No sentence is not clear, please rephrase it.

Author Response

Thanks for your precious suggestions. We have revised the manuscript according to your comments.
